# Chemogenomics for steroid hormone receptors (NR3)
Espen Schallmayer[1], Laura Isigkeit[1], Lewis Elson[1], Susanne Müller [1], Stefan Knapp [1], Julian A. Marschner[2] & Daniel Merk [1,2] ✉

The nine human NR3 nuclear receptors translate steroid hormone signals in transcriptomic responses and operate multiple highly important processes ranging from development over reproductive tissue function to inflammatory and metabolic homeostasis. Although several NR3 ligands such as glucocorticoids are invaluable drugs, this family is only partially explored, for example, in autoimmune diseases and neurodegeneration, but may hold therapeutic potential in new areas. Here we report a chemogenomics (CG) library to reveal elusive effects of NR3 receptor modulation in phenotypic settings. 34 highly annotated and chemically diverse ligands covering all NR3 receptors were selected considering complementary modes of action and activity, selectivity and lack of toxicity. Endoplasmic reticulum stress resolving effects of N3 CG subsets in proof-of-concept application validate suitability of the set to connect phenotypic outcomes with targets and to explore NR3 receptors from a translational perspective.

As endocrine signal molecules, steroid hormones operate a multitude of physiological processes with major phenotypic impact. The stress hormone cortisol, for example, widely affects glucose metabolism and inflammation[1,2], aldosterone maintains cardiovascular homeostasis[3], and the sex steroids (estrogens, androgens and gestagens) regulate biological sex differentiation and reproduction[4]. While several steroid hormones exhibit also rapid non-genomic actions[5,6], the transcriptional effects of corticosteroids and sex steroids are mediated via the nuclear hormone receptors of the NR3 family[7,8] (Fig. 1). NR3 receptors act as ligand-sensing transcription factors and comprise nine proteins with common architecture[9,10] but substantially different ligand recognition.

The pharmacological importance and therapeutic potential of NR3 receptors mediating steroid hormone activities has long been recognized. Invaluable drugs like glucocorticoids for anti-inflammatory treatment[11] and estrogens for contraception[12] have been discovered decades ago and still have remarkable importance. Additionally, targeting of sex steroid receptors (ER, NR3A; AR, NR3C4) is a widely used approach in treating hormone-dependent cancer[13,14] and mineralocorticoid receptor (MR, NR3C2) antagonists are key drugs in cardiovascular diseases[15]. Additionally, a growing body of research supports importance of steroid hormones and their receptors in multiple other organs and pathologies including, for example, immunology and autoimmune diseases[16,17], metabolic dysfunction[18,19], and neurodegeneration[20,21]. Further exploration of NR3 receptors from a translational perspective may, therefore, open new therapeutic avenues, and sets of highly annotated NR3 ligands for chemogenomics (CG)[22–24] could serve as a valuable tool to reveal unprecedented roles of the steroid hormone receptors.

CG is an emerging approach to target identification and validation employing optimized libraries of extensively characterized bioactive molecules for phenotypic screening in disease-relevant in vitro models[22–24]. Only few dedicated CG compound sets have been developed for this purpose, e.g., for kinases[25], and NR1 nuclear hormone receptors[26], but a vast collection of optimized and broadly characterized bioactive molecules for hundreds of targets[27] is available from medicinal chemistry programs to assemble further CG sets and promote translational target identification.

Here we report the rational compilation of a CG library for the NR3 nuclear hormone receptors, its comprehensive characterization, and its application in phenotypic settings. Rigorous filtering of available NR3 modulators and optimization of their combination for chemical diversity and non-overlapping selectivity profiles resulted in a set of 34 compounds fully covering the nine NR3 receptors. Proof-of-concept application of the set indicated involvement of ERR (NR3B) and GR (NR3C1) in regulation and resolution of endoplasmic reticulum stress validating suitability of the assembled set for CG. The publicly available NR3 CG library enables translational exploration of NR3 nuclear hormone receptors and investigation of sex differences in phenotypic settings to unveil elusive potential of this important transcription factor family.

[1]Institute of Pharmaceutical Chemistry, Goethe-Universität Frankfurt, Frankfurt, Germany. [2]Department of Pharmacy, Ludwig-Maximilians-Universität München, Munich, Germany. ✉e-mail: daniel.merk@cup.lmu.de

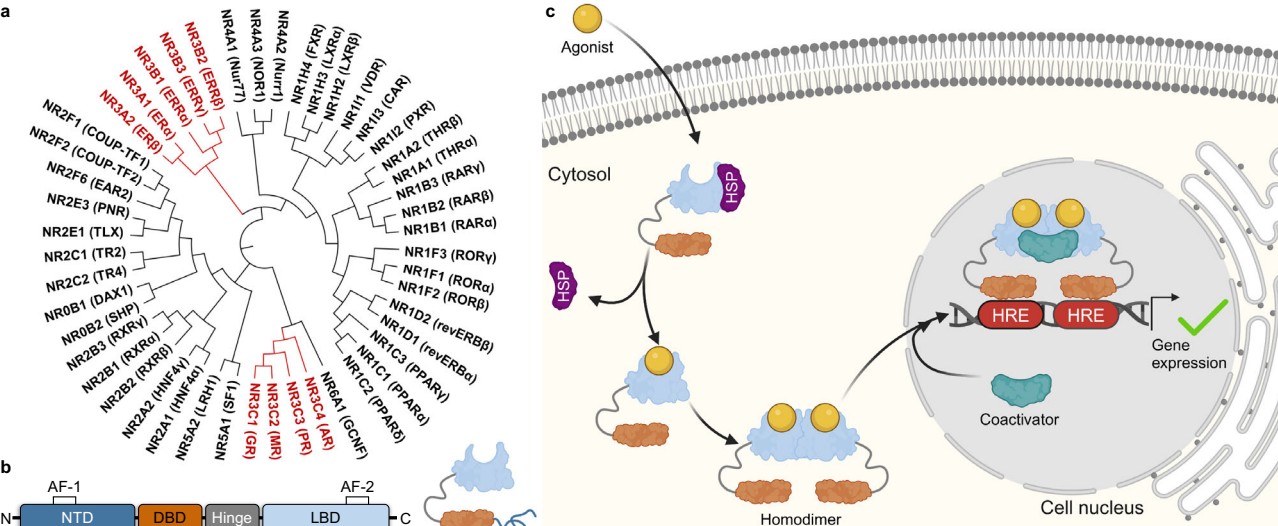

**Fig. 1 | Structure and function of NR3 receptors. a** Phylogenetic tree of the NR family comprising 48 members in human. NR3 receptors are labeled in red. **b** The archetypal domain structure of NRs is composed of an unordered N-terminal domain (NTD; dark blue), containing the ligand-independent activation function 1 (AF-1), a DNA binding domain (DBD; orange) with two zinc finger motifs for binding to hormone response elements (HREs) on the DNA, a flexible hinge region (grey), and a ligand binding domain (LBD; light blue) with the ligand-dependent activation function 2 (AF-2). **c** General activation mechanism of NR3 receptors. Colors corresponding to modular domain structure in (**b**). Before agonist binding (yellow), most NR3 receptors are located in the cytosol bound to heat shock proteins (HSPs; purple) preventing diffusion of the receptor into the nucleus. Agonist binding leads to conformational changes, release of the HSP, and receptor homodimerization. The resulting dimer diffuses to the nucleus, binds the corresponding HRE (red) on the DNA (dark grey), and recruits coactivators (cyan) to initiate gene transcription. Estrogen receptors (ER; NR3A) and progesterone receptor (PR; NR3C3) can also be DNA-bound in the nucleus independently of ligand binding and thus deviate from the general mechanism[72–75]. Created in BioRender. Merk, D. (2025) https://BioRender.com/a87p902.

## Results

### Identification of CG compound candidates for NR3

9361 NR3 ligands ($EC_{50}/IC_{50} \leq 10\ \mu M$) are annotated in public compound/ bioactivity databases (ChEMBL[28], PubChem[29], IUPHAR/BPS[30], BindingDB[31], Probes&Drugs[32]; compiled in ref. 27) with asymmetric distribution over the nine NR3 receptors (Fig. 2a, Supplementary Data 1), reflecting the varying interest of medicinal chemistry in the different targets. Especially the estrogen receptor-related receptors (ERRα-γ, NR3B1-3) are not well covered by potent and selective ligands.

To identify an optimal set of CG compounds for NR3 receptors, the annotated ligands of the protein family were systematically filtered based on the following criteria (Fig. 2b). To enable a broad use of the selected compounds for CG without restrictions, we prioritized commercially available compounds (Supplementary Table 3). Among them, NR3 ligands with a potency $\leq 1\ \mu M$ were considered as candidates with few exceptions for the poorly covered NR3B family, where we also tolerated $\leq 10\ \mu M$ potency. Up to five annotated off-targets were accepted in the initial compound selection. Next, we prioritized chemical diversity in the CG set to add orthogonality as chemically different compounds are less likely to share common (unknown) off-targets. Chemical diversity of the compounds was evaluated based on the pairwise Tanimoto similarity computed on Morgan fingerprints[33] and the candidate combination was optimized for low similarity using a diversity picker. As another level of orthogonality, we included NR3 ligands with diverse modes of action (agonist, antagonist, inverse agonist, modulator, degrader) where available. For NR3 targets with more available ligands complying with these criteria, than desired for the CG set, we eventually considered the validity of the remaining candidates based on the intensity of their use in scientific literature. 40 initial CG compound candidates for NR3 receptors were identified in the filtering process for experimental characterization and validation (Supplementary Table 1) including 15 NR3A ligands, 8 NR3B ligands, and 20 NR3C ligands (considering the respective main target). Of note, some CG compound candidates (e.g., (Z)-4-hydroxytamoxifen or diethylstilbestrol) act on more than one NR3 main target.

### Profiling of CG compound candidates

The selection of 40 CG candidates was acquired from commercial vendors (purity ≥95%) and applied to initial toxicity screening. Cytotoxicity was determined in HEK293T cells considering growth-rate, metabolic activity as well as apoptosis and/or necrosis induction to evaluate suitability of the compounds for CG application in cellular settings (Fig. 3a–d, Supplementary Fig. 1, Supplementary Data 1). The candidates were generally well tolerated at concentrations $>>EC_{50}/IC_{50}$ that we recommend for CG application with only four compounds showing mild toxic effects. Diethylstilbestrol reduced the growth-rate at $3\ \mu M$ but was non-toxic at $0.3\ \mu M$. Additionally, AZD9496, ethinylestradiol and budesonide ($3\ \mu M$) mediated slight apoptosis induction without relevant effects on growth-rate, metabolic activity and necrosis. In terms of cytotoxicity, all candidates thus appeared suitable for inclusion in an NR3 CG set.

To probe selectivity of the candidates in the NR superfamily all compounds were tested for agonistic, antagonistic and inverse agonistic activity in uniform hybrid reporter gene assays[34] on twelve receptors representing the NR1 (THRα, RARα, PPARγ, RORγ, LXRα, VDR, PXR, CAR), NR2 (HNF4α, RXRα), NR4 (Nur77) and NR5 (LRH1) families (Fig. 3e, Supplementary Fig. 2b, Supplementary Data 1). At concentrations $>>EC_{50}/IC_{50}$ for the respective NR3 target, the candidates with exception of biochanin A were favorably selective with few and non-overlapping off-target activities indicating the compounds as suitable for CG.

Next, we screened the candidates for binding to a panel of liability targets by differential scanning fluorimetry (DSF, Fig. 3g, Supplementary Fig. 2c, Supplementary Data 1). This panel comprised ten highly ligandable kinases and bromodomains whose modulation causes strong phenotypes which would compromise use in CG[26]. At $20\ \mu M$ test concentration, the NR3 CG candidates showed only few and weak interactions with the liability target proteins, and, importantly, the candidate sets for NR3 subfamilies had no common off-targets. The liability screening results thus supported suitability of all candidates for CG, too.

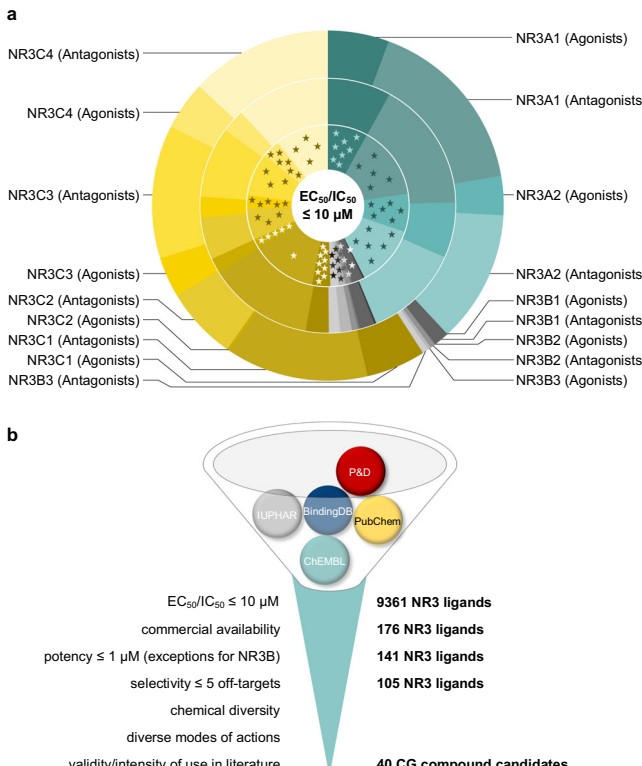

**Fig. 2 | NR3 ligands and selection process. a** Distribution of the 9361 NR3 agonists and antagonists annotated in public repositories over the individual NR3 receptors (outer circle), the 176 commercially available NR3 ligands (middle circle), and the 141 NR3 ligands complying with the potency cutoff (inner circle). Stars represent the selected 40 CG compound candidates (assigned to NR3 subfamilies considering their respective main targets). Compounds and associated data are listed in Supplementary Table 3. **b** Stepwise filtering process for identification of NR3 ligands as CG compound candidates and numbers of remaining candidates for each step.

## Assembly and characteristics of the NR3 CG set

Based on the favorable results from toxicity and selectivity profiling, we assembled a final set of 34 NR3 CG compounds by rational comparison and selection (Figs. 4 and 5). From the 40 evaluated candidates, we removed the ER (NR3A) degrader ARN810 as this mode-of-action was sufficiently represented by AZD9496 and fulvestrant, which have less off-targets than ARN810 (Supplementary Fig. 2, Supplementary Tables 1, 2, Supplementary Data 1). Similarly, we dropped ethinylestradiol in favor of the more selective and less toxic ER (NR3A) agonists (*R,R*)-THC and diarylpropionitrile, the GR/MR (NR3C1/NR3C2) agonist fluticasone propionate in favor of the more potent and selective fludrocortisone acetate, and the NR3C antagonist *Z*-guggulsterone in favor of the more potent and substantially more selective PF-03882845. Additionally, biochanin A and budesonide were removed for their less preferable selectivity profiles within the NR3 family (Supplementary Fig. 2, Supplementary Table 1, 2, Supplementary Data 1).

The final CG set fully covers the NR3 family with 12 NR3A ligands, 7 NR3B ligands and 17 NR3C ligands (Figs. 4a, b, and 5a, b) and includes at least two modes of action with activating and inhibiting ligands for every NR3 subfamily. As intended, the compound collection exhibited high chemical diversity with low pairwise similarity computed on Morgan fingerprints (Fig. 4d) and high scaffold diversity with the 34 compounds representing 29 different skeletons (Fig. 4b).

All selected NR3A and NR3C ligands exhibit sub-micromolar potency on their targets reported in the literature and are suitable for application in CG at 0.3 μM or 1 μM concentration to ensure a full on-target effect with minimal off-target activities. The poorly explored NR3B family for which substantially less ligands are available required inclusion of less potent CG compounds that have to be used at higher concentrations (3 μM or 10 μM)

to achieve a full on-target effect. Recommended concentrations of all selected compounds for application in CG (Fig. 5a, b) have been validated to be non-toxic in cellular setting and modulate the intended NR3 receptor with favorable selectivity within the NR family. Additionally, the combination of several ligands with high chemical diversity in the CG set accounts for unknown off-targets outside the target family which cannot be excluded but likely do not overlap in the chemically diverse ligand collection per target thus enabling deconvolution of phenotypic outcomes observed with the CG library to individual NR3 receptors (Fig. 5c).

## Application of the NR3 CG set

For a proof-of-concept application, we probed the impact of the final NR3 CG set on the inflammatory response to ionomycin/ phorbol-12-myristat-13-acetate (PMA) by observing NF-κB activity in HEK293 cells (Fig. 6a, Supplementary Data 1). GR agonists (GSK9027, AZD5423, beclomethasone, mapracorat, fludrocortisone acetate) consistently delayed and strongly diminished NF-κB activity in line with their anti-inflammatory properties[35,36]. Additionally, an anti-inflammatory effect was evident for AR activation (andarine, BMS564929) while AR antagonists (PF-998425, cyproterone acetate, enzalutamide) tended to enhance NF-κB activity. An opposite trend was observable for ER with inhibitors (AZD9496, fulvestrant and bazedoxifene) reducing the inflammatory response and agonists (diarylpropionitrile, propylpyrazoletriol, ormeloxifene, (*R,R*)-THC) having no impact on NF-κB activity. These results are in line with previous reports on pro-inflammatory effects of ER activation[37,38] and AR agonist mediated anti-inflammatory activity[39,40].

Next, we used the CG set to probe potential NR3-mediated effects in endoplasmic reticulum stress by monitoring activity of the activating transcription factor 6 (ATF6), which is involved in the unfolded protein response and activated in endoplasmic reticulum stress[41–43]. Hela cells treated with tunicamycin to induce endoplasmic reticulum stress displayed markedly enhanced ATF6 activity (Fig. 6b, Supplementary Data 1). ERR agonists (GSK4716, DY131) strongly ameliorated the tunicamycin-induced endoplasmic reticulum stress while the ERR inhibiting CG compounds (XCT790, GSK5182, (*Z*)-4-hydroxytamoxifen (3 μM), diethylstilbestrol) lacked this effect. Consistently weaker ATF6 activity was also evident for GR agonists (GSK9027, AZD5423, beclomethasone, mapracorat, fludrocortisone acetate) but not the antagonist mifepristone. These CG results thus point to an involvement of ERR and GR in endoplasmic reticulum stress regulation and resolution that might be therapeutically explored.

ER modulators, in contrast, displayed inconsistent effects on ATF6 activity. While the ER agonists ormeloxifene, (*R,R*)-THC, and (*Z*)-4-hydroxytamoxifen (0.3 μM) mediated resolution of the tunicamycin-induced endoplasmic reticulum stress, the further agonists propylpyrazoletriol, diarylpropionitrile, WAY20070 and diethylstilbestrol failed to reduce ATF6 activity indicating that the effects of ormeloxifene, (*R,R*)-THC, and (*Z*)-4-hydroxytamoxifen were not ER-mediated. In line with this interpretation, endoplasmic reticulum stress resolving activity has been reported for estrogen in various cell types but this effect was mainly linked to the membrane estrogen receptor GPER[44–47].

The proof-of-concept applications thus demonstrated that the NR3 CG set is suitable to correlate phenotypic outcomes with molecular targets and to generate new biological hypotheses. Despite the vast knowledge on NR3 nuclear hormone receptors, it may enable unprecedented discoveries and open new therapeutic opportunities.

## Discussion

Nuclear receptors of the NR3 family act as cellular steroid hormone sensors and translate endocrine signals into transcriptional responses. They operate innumerable processes and have been identified as highly attractive drug targets which is reflected by glucocorticoids as most widely used strong anti-inflammatory drugs or estrogens as female contraceptives. Despite several success stories of NR3 ligands as drugs, the protein family is not comprehensively explored for its therapeutic potential. With increasing availability

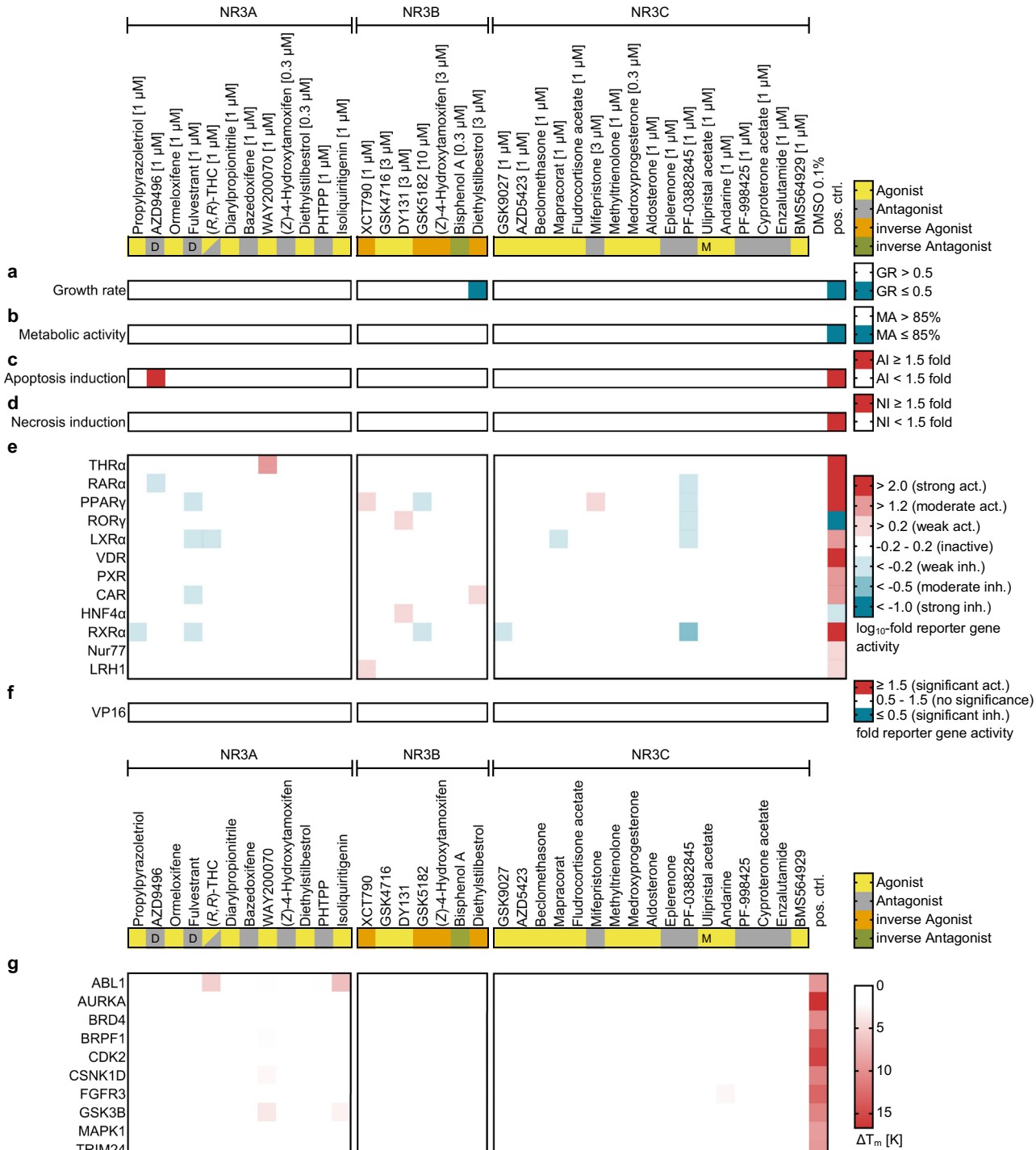

**Fig. 3 | Toxicity and selectivity profiling of the NR3 set.** Compounds are labeled according to their mode of action (yellow - agonist, yellow with label M - modulator, grey - antagonist, grey with label D - degrader, orange - inverse agonist, green - inverse antagonist) and grouped by their NR3 subfamily main targets. **a–d** Toxicity profiling of NR3 CG compound candidates in HEK293T cells considering growth-rate (**a**), metabolic activity (**b**), apoptosis induction (**c**) and necrosis induction (**d**). Cells were incubated with the test compounds at the indicated concentrations solubilized with 0.1% DMSO for 24 h. Growth-rate was determined based on the change in confluence over 24 h normalized to the change in confluence of cells treated with the 0.1% DMSO control ($n = 10$). Metabolic activity was determined with a water-soluble tetrazolium-8 (WST-8) assay ($n = 8$) after 24 h treatment. Apoptotic cells were detected with a fluorogenic DNA dye coupled to the caspase-3/7 recognition sequence (DEVD) after 24 h treatment and the apoptotic cell count was normalized to 0.1% DMSO treated cells ($n = 6$). Necrotic cells were detected with a red fluorescent dye that labels cells with permeable plasma membranes after 24 h treatment and the necrotic cell count was normalized to 0.1% DMSO treated cells ($n = 4$). **e** NR family selectivity profiling of NR3 CG compound candidates in uniform Gal4-hybrid reporter gene assays for selected NR representing the NR1, NR2, NR4 and NR5 families. Reference ligands are listed in the methods section. The heatmap shows NR mediated activation (red; agonists) and inhibition of reporter gene expression (blue; antagonists and inverse agonists) expressed as mean $\log_{10}$ fold reporter activity vs. DMSO ctrl ($n = 3$). **f** Effects of the NR3 CG compound candidates on the ligand-independent transcriptional inducer Gal4-VP16[71,76] to capture non-specific effects on transcriptional activity. The heatmap shows the mean fold Gal4-VP16 dependent reporter gene activity vs. DMSO ctrl ($n = 3$). **g** Profiling of NR3 CG compound candidates for binding to a panel of highly ligandable liability off-targets that mediate strong phenotypes upon inhibition. Compounds were screened at 20 µM by differential scanning fluorimetry (DSF) with the recombinant proteins (2 µM). Positive controls (pos. ctrl) are listed in the methods section. The heatmap shows the mean $\Delta T_m$ calculated by Boltzmann fit ($n = 2$).

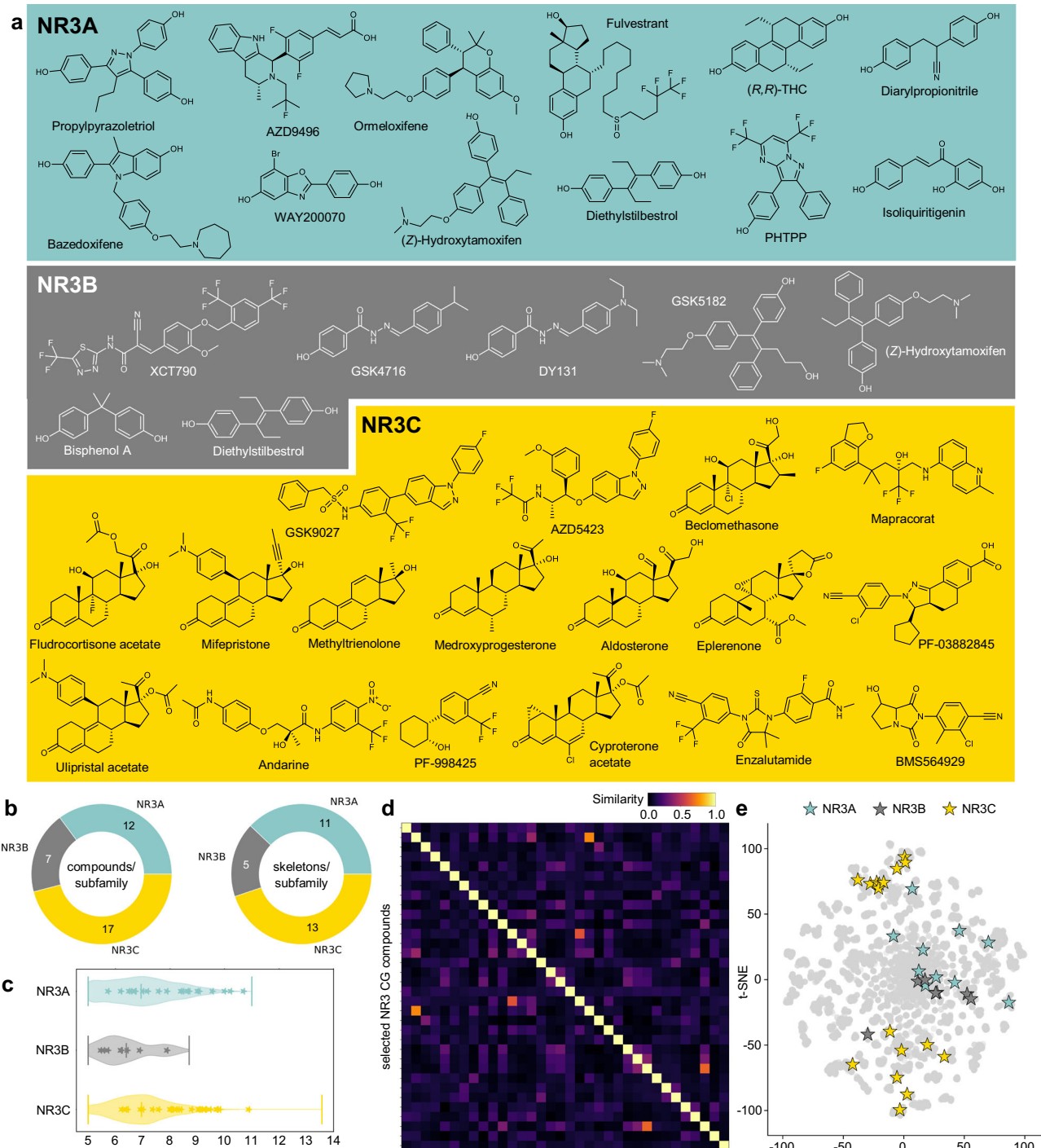

**Fig. 4 | Final set of 34 NR3 CG compounds. a** Chemical structures and most common names of the selected NR3 CG compounds. **b** Pie charts of the number of compounds per subfamily and the number of diverse skeletons per NR3 subfamily. **c** Potency distribution of known NR3 ligands (≤10 µM from dataset[27]) with selected CG compounds highlighted as stars, shown as the negative decadic logarithm of potency. **d** Heatmap of the pairwise Jaccard-Tanimoto similarity of the selected NR3 CG compounds computed on Morgan fingerprints. Axes represent the selected NR3 CG compounds. **e** Distribution of known NR3 ligands (grey, ≤10 µM from dataset[27]) and the selected NR3 CG compounds (highlighted as stars and colored by NR3 subfamilies) visualized by dimensionality reduction with t-distributed stochastic neighbor embedding (t-SNE) computed on Morgan fingerprints. The axes represent the t-SNE dimensions.

of highly sophisticated in vitro models for phenotypic studies on many pathologies[48–51], CG enables systematic exploration of entire protein families for new therapeutic opportunities. For such purposes, we have assembled a CG set covering the NR3 family with carefully selected best-in-class chemical tools.

In CG libraries for target identification and validation, several ligands for one target are combined to account for incomplete selectivity of the utilized compounds[23,26]. Therefore, we have not only considered potency and selectivity as key attributes for chemical tools but also optimized the chemical diversity of the selected compounds to add chemical orthogonality and reduce the probability of common unknown off-targets according to the similarity-property principle[52,53]. Additionally, we focused on commercially available NR3 ligands to enhance accessibility of the set to a broad community.

**Fig. 5 | Characteristics of the selected NR3 CG compounds. a** Recommended concentrations (Recom. conc.) refer to the recommended use of each compound in CG considering potency on the main NR3 target(s), activity type, NR off-targets and known off-targets outside the NR family. p(Potency) values refer to the pEC$_{50}$ for agonists and to pIC$_{50}$ for inverse agonists/antagonists reported in literature (n.a. no data available). Colors denote different NR3 subfamilies and match with the colors in Fig. 4a. Known off-targets of selected NR3 CG compounds are soluble epoxide hydrolase (sEH), phospholipase D1/2 (PLD1/2), aldehyde oxidase (AO), 17β-hydroxysteroid dehydrogenase 1/2 (17β-HSD1/2), nod-like receptor 3 (NLRP3), multi-catalytic protease (MCP), bile salt export pump (BSEP) and several cytochrome P450 (CYP) isoforms. **b** NR3 modulation profiles of the selected CG compounds. Compounds are labeled according to their mode of action (yellow - agonist, yellow with label M - modulator, grey - antagonist, grey with label D - degrader, orange - inverse agonist, green - inverse antagonist) and grouped by their NR3 subfamily main targets. The heatmap shows the potency expressed as pEC$_{50}$ (red; agonists) or pIC$_{50}$ (blue; antagonists and inverse agonists) reported in literature. For a few compounds, only single-point activity data but no EC$_{50}$/IC$_{50}$ have been reported (yellow; n.a.). **c** Correlations between the selected CG compounds and their NR3 targets. Multiple NR3 ligands with diverse activity profiles in the CG set enable deconvolution of phenotypic effects to NR3 receptors.

**a**

| Target family | Compound | Recom. conc. | Main NR target | p(Potency) | Type | NR off-target at recom. conc. | Other off-target at recom. conc. |
|---|---|---|---|---|---|---|---|
| NR3A | Propylpyrazoletriol | 1 µM | 3A1 | 10 | Agonist | | |
| | AZD9496 | 1 µM | 3A1 | 9.6 | Antagonist (Degrader) | 3C3 | |
| | Ormeloxifene | 1 µM | 3A1 | 7.1 | Agonist | | |
| | Fulvestrant | 1 µM | 3A1; 3A2 | 8.5; 8.4 | Antagonist (Degrader) | 1H4; 1I2 | sEH |
| | (R,R)-THC | 1 µM | 3A1; 3A2 | 8.4; 8.4 | Agonist; Antagonist | 1H4 | |
| | Diarylpropionitrile | 1 µM | 3A1; 3A2 | 7.2; 9.1 | Agonist | | |
| | Bazedoxifene | 1 µM | 3A1; 3A2 | 7.6; 7.0 | Antagonist | | |
| | WAY200070 | 1 µM | 3A1; 3A2 | 6.7; 8.7 | Agonist | | |
| | (Z)-4-Hydroxytamoxifen | 0.3 µM | 3A1; 3A2 | 8.6; 9.0 | Antagonist | | PLD1; PLD2 |
| | Diethylstilbestrol | 0.3 µM | 3A1; 3A2 | 10.2; 10.7 | Agonist | | AO |
| | PHTPP | 1 µM | 3A2 | 6.7 | Antagonist | 3A1 | |
| | Isoliquiritigenin | 1 µM | 3A2 | 6.6 | Agonist | 3A1 | 17β-HSD1; 17β-HSD2; NLRP3; MCP |
| NR3B | XCT790 | 1 µM | 3B1 | 6.4 | inverse Agonist | | |
| | GSK4716 | 3 µM | 3B2; 3B3 | n.a.; 5.7 | Agonist | | |
| | DY131 | 3 µM | 3B2; 3B3 | n.a.; 6.9 | Agonist | | |
| | GSK5182 | 10 µM | 3B2; 3B3 | 5.5; 5.6 | inverse Agonist | 3A1 | |
| | (Z)-4-Hydroxytamoxifen | 3 µM | 3B2; 3B3 | 6.2; 6.2 | inverse Agonist | | PLD1; PLD2 |
| | Bisphenol A | 0.3 µM | 3B3 | 7.9 | inverse Antagonist | 3A1; 3A2; 3C4 | |
| | Diethylstilbestrol | 3 µM | 3B3 | 6.2 | inverse Agonist | | AO |
| NR3C | GSK9027 | 1 µM | 3C1 | 8.0 | Agonist | | |
| | AZD5423 | 1 µM | 3C1 | 9.0 | Agonist | 3C2; 3C3 | |
| | Beclomethasone | 1 µM | 3C1 | 8.1 | Agonist | | |
| | Mapracorat | 1 µM | 3C1 | 8.8 | Agonist | | |
| | Fludrocortisone acetate | 1 µM | 3C1; 3C2 | 8.4; 9.7 | Agonist | | |
| | Mifepristone | 3 µM | 3C1; 3C2; 3C3; 3C4 | 9.1; 6.2; 9.7; 8.3 | Antagonist | 3A2 | BSEP; CYP2C8; CYP2C9; CYP3A4 |
| | Methyltrienolone | 1 µM | 3C1; 3C2; 3C3; 3C4 | 8.0; 9.3; 9.3; 10.9 | Agonist | | |
| | Medroxyprogesterone | 0.3 µM | 3C1; 3C3; 3C4 | 8.0; 9.8; 8.2 | Agonist | 3A1 | |
| | Aldosterone | 1 µM | 3C2 | 9.5 | Agonist | | |
| | Eplerenone | 1 µM | 3C2 | 6.9 | Antagonist | | |
| | PF-03882845 | 1 µM | 3C2; 3C3 | 8.0; 6.4 | Antagonist | | |
| | Ulipristal acetate | 1 µM | 3C3 | 7.0 | Agonist (Modulator) | | |
| | Andarine | 1 µM | 3C4 | 8.4 | Agonist | | |
| | PF-998425 | 1 µM | 3C4 | 7.4 | Antagonist | | |
| | Cyproterone acetate | 1 µM | 3C4 | 7.6 | Antagonist | | |
| | Enzalutamide | 1 µM | 3C4 | 6.4 | Antagonist | | |
| | BMS564929 | 1 µM | 3C4 | 9.2 | Agonist | | |

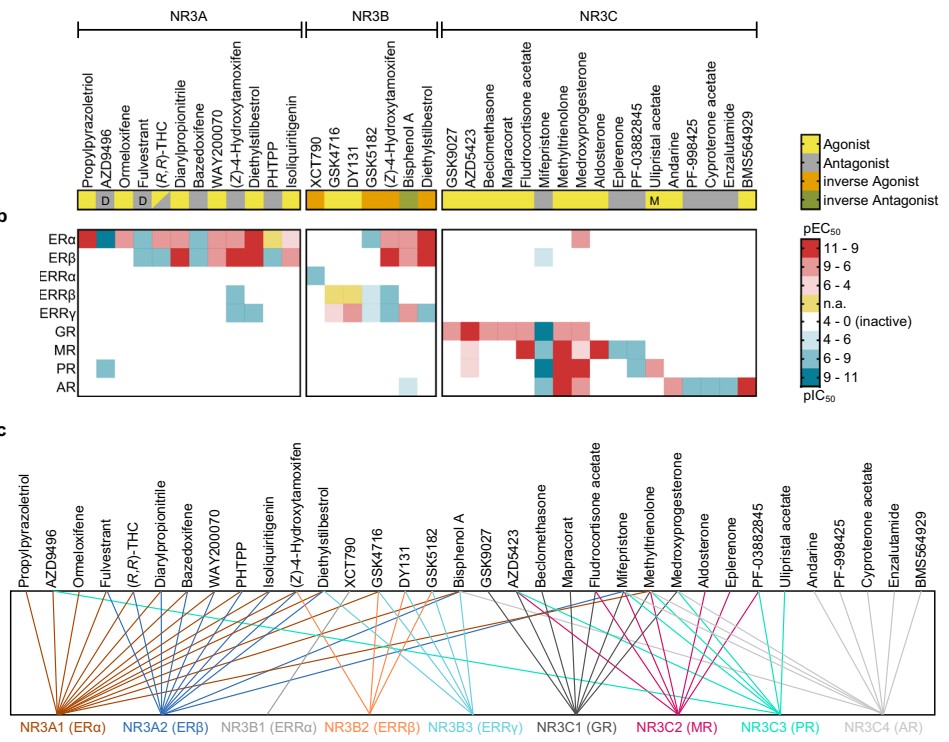

Extensive annotation of chemical tools is essential for their application especially in CG and, therefore, we have broadly tested the CG compound candidates for cytotoxicity, nuclear receptor selectivity, unspecific effects on transcription and interaction with a panel of liability targets with strong phenotypic impact. A few candidates were excluded based on this profiling and the selected compounds revealed clean profiles supporting their confident use in CG. The resulting collection of 34 CG compounds fully covers the NR3 family as a chemical tool to explore these important transcription factors from a translational perspective.

Preliminary application of this NR3 CG set in functional cellular models provided proof-of-concept for its suitability to expose target-effect relationships: The anti-inflammatory GR agonists[35,36] expectedly

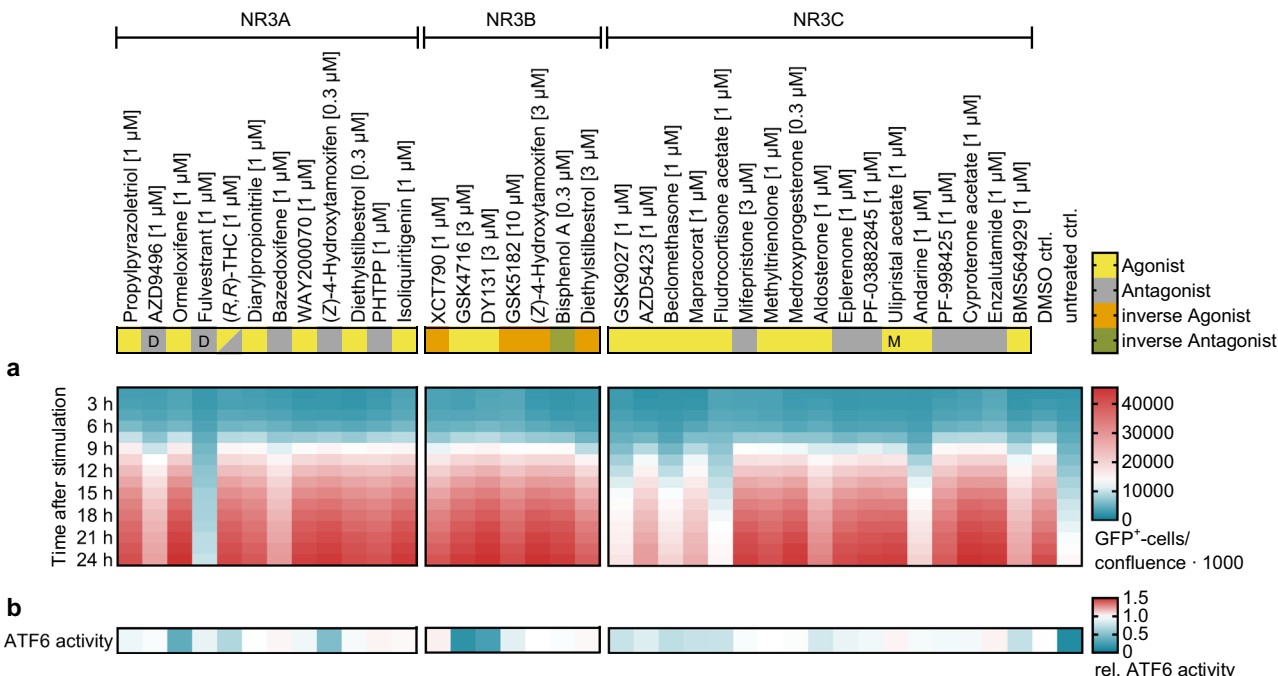

**Fig. 6 | Application of the NR3 CG set.** Compounds are labeled according to their mode of action (yellow - agonist, yellow with label M - modulator, grey - antagonist, grey with label D - degrader, orange - inverse agonist, green - inverse antagonist) and grouped by their NR3 subfamily main targets. **a** Effects of the NR3 CG compounds on NF-κB activity in ionomycin/phorbol-12-myristat-13-acetate (PMA) treated HEK293 NF-κB reporter cells. The heatmap shows the mean number of GFP expressing cells normalized to confluence per well; $n = 3$. **b** Effects of the NR3 CG compounds on ATF6 activity in tunicamycin treated Hela ATF6 reporter cells. The heatmap shows the mean relative ATF6 reporter activity normalized to DMSO ctrl; $n = 5$.

counteracted ionomycin/PMA-induced NF-κB activity in HEK293 cells and our results for NF-κB suppression by ER inhibitors and AR activators support similar observations in previous studies[37–40]. In an ATF6 reporter cell line, application the CG set revealed an impact of several NR3 receptors on endoplasmic reticulum stress. While such activity has been described, e.g., for single ER and GR modulators[54–57], the consistent effects observed for the GR agonist and ERR modulator subsets provide an additional level of confidence for direct involvement of these receptors in regulating endoplasmic stress responses and the pronounced and uniform impact of ERR agonists on ATF6 activity may be worth further evaluation. These results from functional cellular assays showcase the potential of the curated NR3 modulator set to link NR3 receptors to phenotypic effects and encourage further application of the CG approach to this protein family. Phenotypic in vitro models are on the rise and can mimic various tissues and associated pathologies ranging from cancer over infectious diseases and metabolic dysfunction to neurodegeneration[48–51]. Allowing substantially higher throughput than traditional preclinical models, such 3D cellular systems are a perfect match with the concept of CG to explore new target-effect relationships. Application of the NR3 CG set according to this concept in such modern in vitro models may unveil new therapeutic opportunities.

## Methods
### CG compound selection
**Computational methods for CG candidate compound selection.** CG compound candidate selection was performed with Konstanz Information Miner (KNIME, version 4.5)[58] software with RDKit (version 2022.09.1) nodes. Compounds were extracted from a recently compiled dataset[27] containing multiple annotated bioactivities from five public databases (PubChem[29], ChEMBL[28], IUPHAR/BPS[30], BindingDB[31] and Probes&Drugs[32]). The compounds were prefiltered for bioactivity ≤1 μM (NR3A, NR3C) and ≤10 μM (NR3B) on the intended main target based on the published bioactivities. Commercial availability was determined by combining vendor databases (Tocris, Sigma, Cayman

Chemicals, Selleckchem and MedChemExpress) with the compound selection using ChEMBL IDs and SMILES. The number of off-targets was determined by searching the dataset for other targets of the pre-selected compounds with annotated bioactivities at ≤10 μM. Chemical diversity of the CG candidate compounds was evaluated based on Morgan fingerprints[33] calculated using the RDKit Fingerprint node with following settings: Fingerprint type = Morgan, number of bits =1024 and radius =2. For skeleton analysis, RDKit Find Murcko Scaffolds with the setting create frameworks was used. Up to 10 chemically diverse molecules and skeletons were selected for each target using the RDKit Diversity Picker. Bioactivity values annotated in the database were compared/validated with the original literature and the indicated mode of action before candidates were selected for profiling. Chemical property analysis was done using KNIME and python (version 3.8)[59]. The similarity heatmap was generated using Tanimoto similarity[60] computed on Morgan fingerprints. For the t-SNE and potency analysis, all NR3 compounds with annotated bioactivity ≤10 μM were selected from the dataset[27], and the t-SNE was computed based on Morgan fingerprints (from RDKit in KNIME) using scikit-learn (version 1.3.)[61] with the following settings: n_components = 2, perplexity = 30, learning_rate = auto, init = pca.

### Biological quality control
**Evaluation of cytotoxicity.** HEK293T cells (German Collection of Microorganisms and Cell Culture GmbH, DSMZ; #ACC305) were cultured in DMEM high glucose, supplemented with 10% fetal calf serum (FCS), sodium pyruvate (1 mM), penicillin (100 μ/mL) and streptomycin (100 μg/mL) at 37 °C and 5% (v/v) $CO_2$, and seeded in pre-coated (10 μg/mL collagen G solution (Merck KgaA; #L7213) for 30 min at 37 °C) 96-well plates ($2 \times 10^5$ cells per well). After 24 h, the cells were treated with the respective NR3 CG compound candidate at the recommended concentration, bexarotene (100 μM) or doxorubicin (30 μM) as positive controls in DMEM high glucose supplemented with 0.1% FCS, penicillin (100 μ/mL), streptomycin (100 μg/mL) and 0.1%

DMSO, or 0.1% DMSO alone as an untreated control. Each sample was prepared in at least four biologically independent replicates. After 24 h incubation, the medium was aspirated, and the cells were incubated for 30 min with staining medium containing 1 µM NucView® 405 fluorogenic caspase-3 substrate (Biotium, Fremont, California, USA; #10405) and Live-or-Dye Nuc-Fix Red (0.05x, Biotium, Fremont, California, USA; #32010-T) to detect apoptosis and necrosis, respectively. After incubation, a total of six fluorescence images per well at 10x magnification were taken to detect NucView®-positive (Ex: 381−400 nm, Em: 414–450 nm) and Live-or-Dye-positive cells (Ex: 543−566 nm, Em: 580–611 nm) using on a Tecan Spark Cyto (Tecan Group AG, Männedorf, Switzerland). Reference readings for background correction and detection of auto-fluorescence were taken at 414–450 nm prior to staining. Thereafter, the medium was aspirated and the cells were incubated for 3 h with 90 µL DMEM high glucose supplemented with 0.2% FCS, penicillin (100 µ/mL) and streptomycin (100 µg/mL) and additionally containing 10 µL Cell Counting Kit 8 solution (CCK-8; MedChemExpress, Monmouth Junction, New York, USA; #HY-K0301), and absorbance was measured at 1 h, 2 h and 3 h of incubation at 450 nm on a Tecan Spark Cyto to assess metabolic activity of the cells. Before drug administration, after the first medium exchange, 24 h after drug administration, and after fluorescence imaging cell confluence was assessed using the Tecan Spark Cyto, to account for changes in cell confluence due to drug administration and cell handling. Compound effects on cell confluence, apoptosis, metabolic activity and necrosis were normalized to the vehicle control of each biological replicate individually.

**Liability panel screening by DSF.** In-house produced recombinantly expressed proteins[62] of the liability panel targets were diluted in a buffer containing 10 mM HEPES (pH 7.5) and 500 mM NaCl to a concentration of 2 µM. SYPRO Orange (Invitrogen, Carlsbad, California, USA; #S6650) as a fluorescence probe was added at a 1:1000 dilution to the proteins. The protein-dye mixtures were transferred to a 384-well plate with 20 µL volume, and the NR3 CG compound candidates were added at a final concentration of 20 µM using a Echo 525 liquid handler (Labcyte, San Jose, California, USA; #001-10080). Temperature-dependent unfolding profiles were measured on a QuantStudio™ 5 real-time PCR machine (Thermo Fisher Scientific, Waltham, Massachusetts, USA; #A28140). Excitation and emission filters were set to 465 nm and 590 nm, respectively. The temperature was increased 3 °C per minute to a maximum temperature of 85 °C. Experiments were run as technical duplicates. As positive controls (pos. ctrl.) staurosporine (ABL1, AURKA, CDK2, FGFR3 and GSK3B), (+)-JQ1 (BRD4), GSK6853 (BRPF1), PK016714a (CSNK1D), GDC-0994 (MAPK1) and IACS-9571 (TRIM24) were used with 20 µM each. Data were analyzed with the internal Thermal Shift Software (version 1.4, Thermo Fisher Scientific) using the Boltzmann equitation to determine the inflection point of the transition curve. Differences in melting temperature are given as ΔTm in K.

### CG compound profiling

**Gal4 hybrid reporter gene assays.** HEK293T cells (ATCC, Manassas, Virgina, USA; #CRL-3216) were cultured to a maximal confluence of 70–80% in DMEM, high glucose, supplemented with 10% FCS, sodium pyruvate (1 mM), penicillin (100 U/mL), and streptomycin (100 µg/mL) at 37 °C and 5% CO$_2$ (v/v), and seeded in clear 96-well plates ($4 \times 10^4$ cells per well). After 20–24 h, medium was changed to Opti-MEM without supplements and the cells were transiently transfected using Lipofectamine™ LTX/Plus™ Reagents (Invitrogen, Carlsbad, California, USA; #15338100) according to the manufacturer's protocol with pFR-Luc (reporter gene; Stratagene, Agilent Technologies, Santa Clara, California, USA; #219050), pRL-SV40 (control gene for normalization of transfection efficiency and cell growth; Promega, Fitchburg, Wisconsin, USA; #E2231) and one pFA-CMV-hNR-LBD clone coding for the hinge region and ligand binding domain (LBD) of the human NR of interest. Five hours after transfection, the cells were incubated with the NR3 CG compound candidates at the recommended concentrations by changing the medium to Opti-MEM supplemented with penicillin (100 µ/mL) and streptomycin (100 µg/mL) additionally containing 0.1% DMSO and the respective test compound or 0.1% DMSO alone as untreated control. Firefly (reporter) and renilla (control) luminescence were measured after 14–16 h incubation using the Dual-Glo® Luciferase Assay System (Promega, Fitchburg, Wisconsin, USA; #E2980) according to manufacturer's protocol on a Tecan Spark® 10M multimode microplate reader (Tecan Group AG, Männedorf, Switzerland). To obtain relative light units (RLU) firefly luminescence was divided by renilla luminescence and multiplied by 1000. Fold activation was obtained by dividing the mean RLU of a test sample by the mean RLU of the DMSO (0.1%) control. Experiments were performed in singlets in at least three biologically independent repeats. The following pFA-CMV-hNR-LBD clones and reference ligands were used: THRα (pFA-CMV-hTHRα-LBD[63], 1 µM T3), RARα (pFA-CMV-hRARα-LBD[64], 1 µM retinoic acid), PPARγ (pFA-CMV-hPPARγ-LBD[65], 1 µM rosiglitazone), RORγ (pFA-CMV-hRORγ-LBD[66], 1 µM SR1001), LXRα (pFA-CMV-hLXRα-LBD[67], 1 µM T0901317), VDR (pFA-CMV-hVDR-LBD[64], 1 µM calcitriol), PXR (pFA-CMV-hPXR-LBD[64], 1 µM SR12813), CAR (pFA-CMV-hCAR-LBD[64], 1 µM CITCO), HNF4α (pFA-CMV-hHNF4α-LBD[68], 30 µM compound 9 from ref. [68]), RXRα (pFA-CMV-hRXRα-LBD[64], 1 µM bexarotene), Nur77 (pFA-CMV-hNur77-LBD[69], 100 µM amodiaquine), and LRH1 (pFA-CMV-hSF1-LBD[70], in-house compound). pECE-SV40-Gal4-VP16[71] (gift from Lea Sistonen; Addgene, Watertown, Massachusetts, USA; #71728) encoding the ligand-independent transcriptional inducer Gal4-VP16 was used instead of a pFA-CMV-hNR-LBD clone to test for unspecific compound effects on reporter activity.

### Assays for NR3 CG set application

**NF-kB activity assay.** HEK293 cells containing a GFP reporter gene for NF-kB activity (NF-κB Leeporter™ GFP Reporter-HEK293 Cell Line, Abeomics Inc., San Diego, California, USA; #14-700ACL) were cultured in DMEM supplemented with 10% heat-inactivated FCS, penicillin (100 µ/mL), streptomycin (100 µg/mL) and puromycin (3 µg/ml) at 37 °C and 5% (v/v) CO$_2$ and seeded in 96-well plates ($3 \times 10^4$ cells per well). After 24 h, cells were incubated with the test compounds solubilized with 0.1% DMSO in the same medium without puromycin. After 4 h incubation, ionomycin (100 ng/mL) and phorbol-12-myristat-13-acetate (PMA, 10 ng/mL) was added, and the cells were imaged for confluence and GFP expression every 90 min. over 24 h using a Tecan SPARK Cyto (Tecan). The number of GFP expressing cells was normalized to confluence in each sample. Each sample was repeated in three biologically independent experiments.

**Endoplasmic reticulum stress assay.** Hela cells containing a Renilla luciferase reporter gene for ATF6 activity (ATF6 Leeporter™ Luciferase Reporter-HeLa Cell Line, Abeomics Inc.; #14-138ACL) were cultured in DMEM supplemented with 10% heat-inactivated FCS, penicillin (100 µ/mL), streptomycin (100 µg/mL) and puromycin (3 µg/ml) at 37 °C and 5% (v/v) CO$_2$ and seeded in 96-well plates ($1 \times 10^4$ cells per well). After 24 h, cells were incubated with the test compounds solubilized with 0.1% DMSO in the same medium without puromycin. After 4 h incubation, tunicamycin (100 ng/mL) was added. After another 18 h incubation, cells were assayed for renilla luminescence using DualGlo reagent (Promega) on a Tecan SPARK Cyto (Tecan). Test samples were normalized to the 0.1% DMSO control. Each sample was repeated in five biologically independent experiments.

### Reporting summary
Further information on research design is available in the Nature Portfolio Reporting Summary linked to this article.

## Data availability

All data supporting the results of this study are available in the Supplementary Information. Source data are provided with this paper in Supplementary Data 1.

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

## Acknowledgements

This work has received funding from the Innovative Medicines Initiative 2 Joint Undertaking (JU) under grant agreement No. 875510. The JU receives support from the European Union's Horizon 2020 research and innovation program, EFPIA, Ontario Institute for Cancer Research, Royal Institution for the Advancement of Learning McGill University, Kungliga Tekniska Hoegskolan, and Diamond Light Source Limited. This research was co-funded by the European Union (ERC, NeuRoPROBE, 101040355). Views and opinions expressed are however those of the author(s) only and do not necessarily reflect those of the European Union or the European Research Council. Neither the European Union nor the granting authority can be held responsible for them.

## Author contributions

E.S., L.I., and D.M. identified and selected the CG compound candidates. L.I. performed the computational analysis. E.S., L.E., and J.M. performed the in vitro experiments and analyzed the data. E.S. and L.I. visualized the results. S.M., S.K., and D.M. conceived the study. D.M. supervised the research. E.S. and D.M. wrote the manuscript with contributions from all authors.

## Funding

## Competing interests
The authors declare no competing interests.
