## [Transparent Peer Review file · Communications Chemistry]

Chemogenomics for steroid hormone receptors (NR3)

Corresponding Author: Professor Daniel Merk

Version 0:

Reviewer comments:

Reviewer #1

(Remarks to the Author)

The manuscript by Merk and co-workers describes a thorough study on "Chemogenomics for steroid hormone receptors (NR3)".

The authors present a chemogenomics (CG) library designed to uncover subtle effects of NR3 receptor modulation in phenotypic contexts. Their library includes 34 well-characterized and chemically diverse ligands that target all NR3 receptors, selected based on their varied mechanisms of action, activity, selectivity, and lack of toxicity. Proof-of-concept studies demonstrated that the CG set can potentially connect phenotypic outcomes to targets and explore NR3 receptors from a translational perspective.

I think it is fair to say that the authors have performed an extensive amount of work in this manuscript.

Their claims are novel and will definitely be of interest to others in the community especially translational scientists working in the nuclear receptor area.

- The introduction and the results sections are very well written and provide thorough explanation on how the set was assembled, characterised and used.
- Figures are well paired with the text and provide sufficient information for the reader to understand the experiment and the corresponding results.
- The SI section is also very well curated with clear tables and references.
- It would be useful to expand a little bit more on how the authors believe this CG set can be used by others in the field apart from inflammatory response and endoplasmic reticulum stress along with any possible limitations.

Upon thorough review of the comments provided above, I suggest publication in Communications Chemistry.

Reviewer #2

(Remarks to the Author)

In this manuscript, the authors report on the compilation of a chemogenomics library of commercially available ligands for the NR3 nuclear hormone receptor family and they use of this library in phenotypic settings.

The authors start off with a well considered selection of 40 compounds and their general profiling regarding toxicity, off-target activity etc. This resulted in a final set of 34 compounds. Figures 4 and 5 and connected text then provide a more in depth overview of the molecular and bioactive properties of the compound set.

In the final part of the manuscript (centering on Figure 6), the authors performed two profiling experiments of their library on two signaling pathways in two different cell lines.

As a whole, this study reports on an interesting chemogenomics library, it thorough characterization and first initial illustration of potential value to the broader field. It can be assumed that with this study, there will be quite some interest generated by biomolecular scientists to take on this library into their studies.

I see no major issues that need to be addressed, but would suggest the authors to take the minor suggestions listed below into account.

Minor details:

-Figure 1c. While the activation mechanism generally reflects that of the NR3 class, it should be noted that for example the ERs in most cells are typically already in the nucleus and chromatin bound, even before ligand stimulation. While this will be too complex to highlight in a simple drawing, maybe the wording of the legend can be tuned to highlight the generality of the mechanism, but also the existence of deviations thereof.

-The statement "Despite their remarkable success as therapeutic targets and strong evidence for unexplored potential, NR3

receptors have not been systematically explored from a translational perspective” seems to be an oversimplification of matters. As also stated by the authors, there are quite a few reports on specific translational aspects of NR3 receptors. These also include ER/AR in other cancers, ERbeta in prostate, GR in several cancers etc. Indeed, there is no unifying study of the translational perspective of these NR3 receptors, but also this current study only touches on some elements of that. It might be wise to formulate carefully regarding this matter to be appreciative of the field.

-The left circle in Figure 2a is somewhat superfluous (same information is also present in right circle).

-It would be good if the 40 selected compounds would be highlighted where they fall in the different family members by positioning them in the right circle of Figure 2a. This will give a direct appreciation of the distribution of the library. For example it will allow the reader to appreciate whether observations regarding specific NR3 family members are based on a single ligand or on a broader chemical representation. It would especially nice, if the ant/ago profile of these ligands could be taken along in this figure. Again this will allow for a much broader appreciation of the data (only an antagonist for a specific receptor class will for example be much less informative than having diverse types of agonism profiles).

-The discussion regarding the role of the different NR3 family members involved in the cell biological studies and observations reported in Figure 6 could be strengthened. Especially the first experiment regarding NF- κ B has very little data interpretation and literature setting.

Reviewer #3

(Remarks to the Author)

This article explores the establishment and application of a chemogenomics (CG) library targeting NR 3 class nuclear receptors. The research team, following meticulous selection and optimization, successfully constructed a CG library composed of 34 highly annotated and chemically diverse ligands that comprehensively encompass all receptors of the NR 3 family. Through a succession of in vitro experiments, the study verified the activity and selectivity of these ligands in cellular models, particularly their application in the regulation of endoplasmic reticulum stress, thereby demonstrating their remarkable potential.

The novelty of the study lies in its systematic approach, which not only takes into account the efficacy and selectivity of the ligands but also optimizes chemical diversity to minimize potential unknown off-target effects. Additionally, the research offers detailed compound screening and biological activity testing methods, providing solid data support for subsequent drug discovery and target validation.

Nevertheless, the discussion section of the article is rather ambiguous in depicting the potential and application prospects of the CG library in drug discovery. It is proposed that the authors could furnish more specific examples or data to substantiate their viewpoints. For instance, they could showcase the application of the CG library in specific disease models and how it reveals new drug targets or therapeutic approaches.

Overall, this article presents a novel perspective and tools for the study of NR 3 class nuclear receptors, which is of great significance for promoting research in this field. It is recommended for publication, with the anticipation that the authors will further validate and refine the application of the CG library in subsequent research.

Here are a few specific suggestions:

1. In the results part, at the end of the second paragraph of "Identification of CG compound candidates for NR3" , it is written that "40 CG compound candidates for NR3 receptors were identified in the filtering process for experimental characterization and validation. Among them were 17 NR3A ligands, 8 NR3B ligands, and 20 NR3C ligands (considering the respective main target).", which differs from the 40 compounds mentioned in the previous sentence. Please explain why the other five compounds were excluded.

2. In Fig. 3 a-e, Fig. 5b, Fig. 6a-b, different types of NR3 receptors (NR3A, NR3B, NR3C) can be marked on different compounds for a clearer view of the ligand-compound relationships for each receptor.

3. In Fig. 4 d-e, the horizontal and vertical coordinates are not clear, more detail can be added to make it easier to understand.

4. Check the uniform format of the reference carefully, such as the abbreviation of the journal name, page number, etc. For example, in ref 45-47, the name of the magazine is not abbreviated, and so on.

Version 1:

Reviewer comments:

Reviewer #1

(Remarks to the Author)

The authors have addressed all the comments from the 3 reviewers and the manuscript has improved to a great extent. I suggest publication in Communications Chemistry in its current form.

Reviewer #2

(Remarks to the Author)

The authors addressed all suggestions (which were minor to begin with) of the reviewers (including mine) convincingly. Manuscript is thus recommended for publication.

Reviewer #3

(Remarks to the Author)

Dear Editor,

Having thoroughly assessed the revised manuscript titled "Chemogenomics for steroid hormone receptors (NR3) (COMMSCHEM-24-0548A)" submitted to Communications Chemistry, and considering the original version along with the authors' detailed responses to the feedback received, I think that the manuscript has been improved and now meets the standards of your esteemed journal.

The authors have addressed all the concerns raised by the reviewers in a thorough manner.

They have expanded on the potential applications of the chemogenomics set beyond inflammatory response and endoplasmic reticulum stress, as well as discussed possible limitations.

They have made changes to the Figures and text, strengthening the discussion of the cell biological studies and observations reported in Figure 6 with more data interpretation and literature references.

From a translational perspective, the manuscript presents a novel and extensive chemogenomics library for the NR3 nuclear hormone receptor family, which is of great interest to the scientific community, particularly translational scientists working in the nuclear receptor area.

I recommend that the manuscript be accepted for publication in your journal.

Thank you for considering my recommendation.

Dear Reviewers

Thank you very much for evaluating our manuscript on a chemogenomics set for the NR3 nuclear hormone receptor family, for your positive feedback and for your constructive input. Your comments were very valuable for improving the manuscript and we have addressed them in full as outlined below. We hope the revised version of the manuscript meets your expectations and we thank you for your further consideration.

Sincerely

Daniel Merk

Reviewer #1 (Remarks to the Author):

The manuscript by Merk and co-workers describes a thorough study on “Chemogenomics for steroid hormone receptors (NR3)”.

The authors present a chemogenomics (CG) library designed to uncover subtle effects of NR3 receptor modulation in phenotypic contexts. Their library includes 34 well-characterized and chemically diverse ligands that target all NR3 receptors, selected based on their varied mechanisms of action, activity, selectivity, and lack of toxicity. Proof-of-concept studies demonstrated that the CG set can potentially connect phenotypic outcomes to targets and explore NR3 receptors from a translational perspective.

I think it is fair to say that the authors have performed an extensive amount of work in this manuscript.

Their claims are novel and will definitely be of interest to others in the community especially translational scientists working in the nuclear receptor area.

- The introduction and the results sections are very well written and provide thorough explanation on how the set was assembled, characterised and used.
- Figures are well paired with the text and provide sufficient information for the reader to understand the experiment and the corresponding results.
- The SI section is also very well curated with clear tables and references.

We thank the Reviewer very much for the careful evaluation of our manuscript and this encouraging feedback.

- It would be useful to expand a little bit more on how the authors believe this CG set can be used by others in the field apart from inflammatory response and endoplasmic reticulum stress along with any possible limitations.

We thank the Reviewer for this remark. We have added further discussion, how the set can be used and how it could reveal new opportunities for NR3 modulation as therapeutic concept, as suggested.

Upon thorough review of the comments provided above, I suggest publication in Communications Chemistry.

We thank the Reviewer for supporting publication of our study in Communications Chemistry.

Reviewer #2 (Remarks to the Author):

In this manuscript, the authors report on the compilation of a chemogenomics library of commercially available ligands for the NR3 nuclear hormone receptor family and they use of this library in phenotypic settings.

The authors start off with a well considered selection of 40 compounds and their general profiling regarding toxicity, off-target activity etc. This resulted in a final set of 34 compounds. Figures 4 and 5 and connected text then provide a more in depth overview of the molecular and bioactive properties of the compound set.

In the final part of the manuscript (centering on Figure 6), the authors performed two profiling experiments of their library on two signaling pathways in two different cell lines.

As a whole, this study reports on an interesting chemogenomics library, it thorough characterization and first initial illustration of potential value to the broader field. It can be assumed that with this study, there will be quite some interest generated by biomolecular scientists to take on this library into their studies.

I see no major issues that need to be addressed, but would suggest the authors to take the minor suggestions listed below into account.

We thank the Reviewer very much for evaluating our manuscript, for the positive feedback and the constructive input. We have addressed all comments and point-by-point answers are given below.

Minor details:

-Figure 1c. While the activation mechanism generally reflects that of the NR3 class, it should be noted that for example the ERs in most cells are typically already in the nucleus and chromatin bound, even before ligand stimulation. While this will be too complex to highlight in a simple drawing, maybe the wording of the legend can be tuned to highlight the generality of the mechanism, but also the existence of deviations thereof.

We thank the Reviewer for this comment. As suggested, we have extended the description/caption of Fig. 1c to reflect this point.

-The statement “Despite their remarkable success as therapeutic targets and strong evidence for unexplored potential, NR3 receptors have not been systematically explored from a translational perspective” seems to be an oversimplification of matters. As also stated by the authors, there are quite a few reports on specific translational aspects of NR3 receptors. These also include ER/AR in other cancers, ERbeta in prostate, GR in several cancers etc. Indeed, there is no unifying study of the translational perspective of these NR3 receptors, but also this current study only touches on some elements of that. It might be wise to formulate carefully regarding this matter to be appreciative of the field.

We thank the Reviewer very much for raising this point. This statement was indeed not well phrased and misleading. We have reworded the respective paragraph.

-The left circle in Figure 2a is somewhat superfluous (same information is also present in right circle).

We thank the Reviewer for this remark and agree that there was some redundancy in Fig. 2. We have redesigned the figure to remove redundancy, add additional information (see below) and improve readability.

-It would be good if the 40 selected compounds would be highlighted where they fall in the different family members by positioning them in the right circle of Figure 2a. This will give a direct appreciation of the distribution of the library. For example it will allow the reader to appreciate

whether observations regarding specific NR3 family members are based on a single ligand or on a broader chemical representation. It would especially nice, if the ant/ago profile of these ligands could be taken along in this figure. Again this will allow for a much broader appreciation of the data (only an antagonist for a specific receptor class will for example be much less informative than having diverse types of agonism profiles).

We thank the Reviewer for these valuable suggestions. We have revised Figure 2 to show the distribution of the candidate compounds over the subfamilies. Additionally, we have revised Figures 3, 5b and 6 to show the activity profile and subfamily distribution of the selected CG compounds. We think the figures are better readable now and thank the Reviewer for this valuable input.

-The discussion regarding the role of the different NR3 family members involved in the cell biological studies and observations reported in Figure 6 could be strengthened. Especially the first experiment regarding NF- κ B has very little data interpretation and literature setting.

We thank the Reviewer for this remark. We agree that discussion on the proof-of-concept applications was scarce. We have extended the discussion of the NF κ B and ATF6 activity assay results and added further literature references to previous observations in this context.

Reviewer #3 (Remarks to the Author):

This article explores the establishment and application of a chemogenomics (CG) library targeting NR 3 class nuclear receptors. The research team, following meticulous selection and optimization, successfully constructed a CG library composed of 34 highly annotated and chemically diverse ligands that comprehensively encompass all receptors of the NR 3 family. Through a succession of in vitro experiments, the study verified the activity and selectivity of these ligands in cellular models, particularly their application in the regulation of endoplasmic reticulum stress, thereby demonstrating their remarkable potential.

The novelty of the study lies in its systematic approach, which not only takes into account the efficacy and selectivity of the ligands but also optimizes chemical diversity to minimize potential unknown off-target effects. Additionally, the research offers detailed compound screening and biological activity testing methods, providing solid data support for subsequent drug discovery and target validation.

Nevertheless, the discussion section of the article is rather ambiguous in depicting the potential and application prospects of the CG library in drug discovery. It is proposed that the authors could furnish more specific examples or data to substantiate their viewpoints. For instance, they could showcase the application of the CG library in specific disease models and how it reveals new drug targets or therapeutic approaches.

Overall, this article presents a novel perspective and tools for the study of NR 3 class nuclear receptors, which is of great significance for promoting research in this field. It is recommended for publication, with the anticipation that the authors will further validate and refine the application of the CG library in subsequent research.

We thank the Reviewer very much for performing peer-review of our manuscript on an NR3 CG set and for the encouraging feedback. The Reviewer's comments were very constructive and helpful. We have addressed them as outlined below.

As suggested, we have extended the discussion, how the set could be used in future applications in modern 3D in vitro phenotypic models. We refrain from naming specific diseases, as we think that the set can be broadly applied without bias.

Here are a few specific suggestions:

1. In the results part, at the end of the second paragraph of "Identification of CG compound candidates for NR3" , it is written that "40 CG compound candidates for NR3 receptors were identified in the filtering process for experimental characterization and validation. Among them were 17 NR3A ligands, 8 NR3B ligands, and 20 NR3C ligands (considering the respective main target).", which differs from the 40 compounds mentioned in the previous sentence. Please explain why the other five compounds were excluded.

We thank the Reviewer for this important remark. The reasons for exclusion were already discussed in the text (section *Assembly and characteristics of the NR3 CG set*). The underlying data leading to exclusion are provided in the Supplementary Information (Suppl. Figs. 1-3) and the reasons for exclusion are also listed in Suppl. Tab. 1. We have added additional reference to these SI resources in the main text. Additionally, we have updated the main text to clarify the compound numbers.

2. In Fig. 3 a-e, Fig. 5b, Fig. 6a-b, different types of NR3 receptors (NR3A, NR3B, NR3C) can be marked on different compounds for a clearer view of the ligand-compound relationships for each receptor.

We thank the Reviewer for this comment. We have revised Figures 3, 5b and 6 to better show the activity types and subfamily distribution of the CG compounds. We think this valuable suggestion has improved readability of these figures.

3. In Fig. 4 d-e, the horizontal and vertical coordinates are not clear, more detail can be added to make it easier to understand.

We thank the Reviewer for this comment. The axes of Figure 4e were indeed not accurately labeled and Figure 4e was not well explained in the caption. We have correctly labeled the axes and improved explanation for Figure 4e, and added explanation of the axes for Figure 4d.

4. Check the uniform format of the reference carefully, such as the abbreviation of the journal name, page number, etc. For example, in ref 45-47, the name of the magazine is not abbreviated, and so on.

We thank the Reviewer for this important comment. Some references were indeed not formatted correctly. All references have been checked and corrected where necessary.